# WEAKLY SUPERVISED MOTION LEARNING FOR CO-SPEECH GESTURE VIDEO GENERATION

## ABSTRACT

Co-speech gesture video generation is fundamental to natural human communication and plays a crucial role in human-computer interaction. Existing approaches typically rely on a two-stage framework, first generating intermediate pose representations before synthesizing the final video. While effective, these methods require extensive pose annotations, which often introduce labeling errors, and still struggle with fine-grained details, particularly in hand generation. To address these challenges, we propose a weakly supervised motion learning framework for co-speech gesture video generation that leverages only audio and video data. Our approach consists of three key stages: (1) a motion encoder that learns a generalizable motion representation from video without pose supervision, (2) a dual-tower architecture that aligns audio with the learned motion representation using an invertible feature extractor, and (3) a video diffusion model that refines fine-grained visual details. During sampling, we introduce a hand refinement method based on initial noise optimization, where learnable noise parameters are optimized via policy gradient to enhance hand synthesis. Extensive experiments on our collected dataset demonstrate that our approach outperforms prior methods across multiple metrics, achieving superior motion fidelity, gesture realism, and overall video quality.

## 1 INTRODUCTION

Co-speech gestures are indispensable to human communication, amplifying meaning, reinforcing intent, and fostering social connection. Generating natural and speech-synchronized gestures is essential for creating lifelike virtual agents and immersive human-computer interactions. Effective solutions could transform fields such as education, assistive technology, and remote collaboration while advancing foundational AI challenges in human-centric synthesis. Co-speech gesture video generation is key to building truly empathetic, interactive artificial systems by bridging the gap between verbal and nonverbal expression.

Previous works typically approach co-speech gesture video generation using a two-stage framework. For example, Vlogger (Corona et al., 2024) first trains an audio-to-motion model to generate dense pose images, followed by a temporal diffusion model to synthesize videos guided by these pose images. Similarly, MYA (Huang et al., 2024) follows this pipeline but uses mesh images instead. S2G (He et al., 2024) first employs an audio-to-keypoint diffusion model, then maps keypoint movements to RGB space via a nonlinear thin-plate spline transformation. While these methods achieve good performance, they rely on extensive dataset annotations, leading to labeling errors that degrade overall quality. Furthermore, they struggle with generating fine-grained details, particularly in generating hands accurately. Additionally, pose annotations are highly sensitive to positional variations, often requiring further alignment during training (Jin et al., 2024).

To this end, we propose a weakly supervised motion learning framework for co-speech gesture video generation that relies solely on video and audio as inputs. We argue that motion information is inherently encoded in video, making additional pose annotations unnecessary, especially for audio-driven tasks. Inspired by recent work in motion transfer, ReenactAnything (Kansy et al., 2024), which trains a single motion embedding per video using diffusion loss and achieves automatic spatial alignment, we extend this idea to improve efficiency. Instead of adapting a single embedding for each video, we introduce a motion encoder to learn a more generalizable motion representation.

Figure 1: During training, previous methods (Corona et al., 2024; Huang et al., 2024; He et al., 2024; Li et al., 2025) often require additional pose annotations, which are time-consuming and prone to labeling errors. In contrast, our approach relies solely on audio and video data, eliminating the need for pose supervision.

Utilizing the trained motion encoder for inference requires an audio-to-motion mapping due to the audio-driven nature of the task. Previous approaches (He et al., 2024; Corona et al., 2024; Huang et al., 2024) rely on audio-to-motion generation, but this is challenging for our implicit motion representation, unlike explicit pose-based methods. Another common approach, audio-to-motion retrieval (Liu et al., 2024a; Zhou et al., 2022), selects motion based on similarity from a database but is computationally expensive, requiring similarity calculations and ranking during inference. To address this, we propose a dual-tower architecture with an audio encoder for processing audio signals and an invertible feature extractor for capturing motion representations. By leveraging the inverse process of the invertible feature extractor during inference, our method directly maps audio inputs to the learned motion space of the motion encoder, eliminating the need for retrieval. Finally, we train the video diffusion model to refine fine-grained visual details.

Additionally, hand generation remains a challenging task. Previous methods (Zhang et al., 2024b; Zhou et al., 2024) rely on pose annotations during training, whereas our approach eliminates the need for such supervision. To address this, we introduce a hand refinement method based on initial noise optimization during sampling. Specifically, we treat $\mu$ and $\sigma$ as learnable parameters and optimize them using a policy gradient approach. This enables the model to sample improved initial noise from the optimized distribution, leading to enhanced hand generation quality.

Our contributions can be summarized as follows: 1) We propose a weakly supervised motion learning framework for co-speech gesture video generation that does not require pose supervision during training. 2) Our framework consists of three key stages: (a) training a generalizable motion encoder, (b) introducing a dual-tower architecture with an audio encoder and an invertible feature extractor to align audio and motion, and (c) training the video diffusion model to enhance fine-grained details. 3) We introduce a hand refinement strategy based on policy gradient optimization, where initial noise is optimized to improve hand synthesis during sampling. 4) Experimental results on our collected dataset demonstrate the effectiveness of our method, achieving superior performance compared to prior approaches.

## 2 RELATED WORK

### 2.1 CO-SPEECH GESTURE VIDEO GENERATION

Co-speech gesture video generation (Mahapatra et al., 2024; Li et al., 2025) is often approached in two stages: generating motion from audio and synthesizing video from that motion. Speech2Gesture (Ginosar et al., 2019) employs a GAN to create 2D skeleton movements, followed by another GAN to produce videos. Speech-Drives-Templates (Qian et al., 2021) uses a VAE in the motion generation stage and applies image warping for video synthesis. Vlogger (Corona et al., 2024) leverages two diffusion models to generate pose images as robust visual controls and the corresponding human videos. MYA (Huang et al., 2024), initially developed for pose-image-guided video generation, can be adapted for co-speech gesture video generation by incorporating an audio-to-motion module (Yi et al., 2023; Liu et al., 2024b; Chen et al., 2024b; Liu et al., 2022b). The same applies to other pose-guided methods (Guan et al., 2024; Yang et al., 2024). S2G (He et al., 2024) utilizes a diffusion model to map audio to keypoint movements, employing a nonlinear thin-plate spline (TPS) transformation to separate latent motion dynamics from video content. ANGIE (Liu et al., 2022a) and DiffTed (Hogue et al., 2024) also rely on similar intermediate motion representations. Alternatively, TANGO (Liu et al., 2024a) employs audio and motion representation learning to retrieve and interpolate motions matching the input audio.

Despite their strong performance, these methods heavily depend on extensive annotation processes or pre-trained pose estimators, which inevitably introduce annotation errors. Moreover, the estimated poses often lack detailed textures and are sensitive to position variations. To address these limitations, we propose a weakly supervised motion learning framework that eliminates the need for labor-intensive annotations.

## 2.2 Zero-shot Audio-driven Video Generation

Zero-shot audio-driven video generation starts with talking-face synthesis (Wang et al., 2021; Peng et al., 2024; Ye et al., 2024; Xu et al., 2024b; Tian et al., 2024b; Xu et al., 2024a), and gradually expands to include hand motion (Meng et al., 2024; Tian et al., 2024a; Guan et al., 2025; Lin et al., 2025b;a). Compared to co-speech gesture video generation, which typically requires hours of video per identity to capture and reproduce an individual's unique gesture style, zero-shot methods focus on generalization across a large number of identities, using only seconds of video per person. While co-speech methods generate diverse, identity-consistent gestures, zero-shot approaches primarily focus on lip-sync and produce only limited hand motion.

We also include a comparison with the open-sourced EchoMimicV2 (Meng et al., 2024) to demonstrate the effectiveness of our method, although it targets a different task.

## 2.3 Motion Transfer

Motion transfer extracts motion from a reference video and applies it to a target, preserving the appearance of the target video while mimicking the movement of the reference video (Yin et al., 2024; Park et al., 2024; Materzynska et al., 2023; Wu et al., 2023b; Zhang et al., 2023b; Li et al., 2024). Several methods have been proposed to address motion transfer. MotionClone (Ling et al., 2024) simplifies motion transfer by using temporal attention as a motion representation. ReenactAnything (Kansy et al., 2024) introduces a trainable motion embedding to enhance motion transfer. Other methods are described in **Appendix**.

Our work is inspired by ReenactAnything (Kansy et al., 2024) due to its automatic spatial alignment capability and seamless integration with existing pretrained diffusion models. However, ReenactAnything learns a single motion embedding for each reference video, whereas our method extends this approach by introducing a generalizable motion encoder.

## 2.4 Hand Refinement

The intricate structure of the human hand, along with frequent finger occlusions and high variability, makes natural hand generation particularly challenging. Several approaches have been proposed to address this issue. In text-to-image generation, some methods first train an anomaly detector, which is then used to guide model fine-tuning (Wang et al., 2024a; 2025) or applied in post-processing through inpainting (Wang et al., 2024c; Fang et al., 2024) and additional control conditions (Lu et al., 2023; Qin et al., 2024) to refine abnormal areas. Other approaches focus on collecting specialized hand datasets and designing tailored training pipelines (Chen et al., 2024c; Zhang et al., 2024a; Zhu et al., 2024; Pelykh et al., 2024; Narasimhaswamy et al., 2024). However, these methods are primarily developed for the image domain and require additional training, fine-tuning, extra control conditions, or inpainting for effective hand generation.

In video generation, previous works (Zhang et al., 2024b; Zhou et al., 2024) incorporate pose information to improve hand generation during training. However, these approaches require extra pose annotations during training and additional model parameters to learn pose representations. In contrast, our proposed refinement method optimizes the initial noise during sampling, eliminating the need for extra annotations and parameters.

## 3 Method

### 3.1 Preliminaries

#### 3.1.1 Latent Diffusion Models (LDMs)

Latent Diffusion Models (LDMs) (Rombach et al., 2022; Blattmann et al., 2023b) generate high-quality images and videos efficiently by operating in a compressed latent space derived from a pre-trained VAE, reducing the computational cost compared to pixel-based methods(Ramesh et al., 2022; Song et al., 2021; Ho et al., 2022). Given an input video $x$, the VAE encoder $\mathcal{E}$ maps it to a latent representation $z = \mathcal{E}(x)$, which is later reconstructed by the decoder $\mathcal{D}$ as $\bar{x} = \mathcal{D}(z)$.

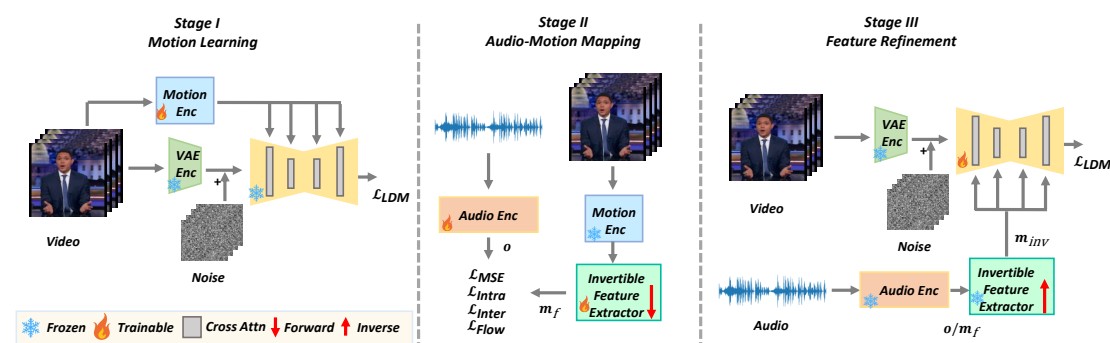

Figure 2: Network Pipeline. In Stage 1, given only video input, we train the motion encoder to learn a general motion representation. In Stage 2, we freeze the motion encoder and train the forward process of the invertible feature extractor along with the audio encoder to learn the audio-motion mapping between $o$ and $m_f$. In Stage 3, given an audio input, we first obtain the audio/motion embedding $o/m_f$ and apply the inverse process of the invertible feature extractor to derive $m_{inv}$, which is then used to train the video diffusion model for feature refinement and detail enhancement via cross-attention. During inference, the process follows Stage 3, where only the first frame and audio serve as inputs. Notably, the appearance features in Stages 1, 3, and inference are controlled by the VAE and CLIP embeddings extracted from the first frame. These appearance feature embeddings are fused with motion information as inputs to the cross-attention mechanism. Further details can be found in I2VGen-XL (Zhang et al., 2023a).

LDMs involve two stages: diffusion gradually adds noise to $z$ over $T$ steps, resulting in $z_t$, while denoising uses a trained model $\epsilon_\theta(z_t, t, c)$ to iteratively remove noise and recover $z_0$. The loss minimizes the noise prediction error: $\mathcal{L}_{\text{LDM}} = \mathbb{E}_{z, \epsilon \sim \mathcal{N}(0,1), t}\left[\|\epsilon - \epsilon_\theta(z_t, t, c)\|_2^2\right]$, where $c$ is optional conditioning, such as text.

We build on I2VGen-XL (Zhang et al., 2023a), which excels at generating complex motion sequences from a single input image (Xing et al., 2023; Lin et al., 2024), while ensuring compatibility with other models like SVD (Blattmann et al., 2023a).

### 3.1.2 COUPLING-BASED NORMALIZING FLOWS

Normalizing Flows (NFs) (Kingma & Dhariwal, 2018; Dinh et al., 2014; 2016) use a series of invertible transformations $f$ to map data $\mathbf{x}$ to latent variables $\mathbf{z}$: $\mathbf{z} = f(\mathbf{x})$, $\mathbf{x} = f^{-1}(\mathbf{z})$. The density of $\mathbf{x}$ can be computed using the change of variables formula: $p(\mathbf{x}) = p(\mathbf{z})\left|\det \frac{\partial f(\mathbf{x})}{\partial \mathbf{x}}\right|$.

Coupling-based NFs (Kingma & Dhariwal, 2018; Dinh et al., 2014; 2016) often use coupling layers for the invertible transformation. Each coupling layer splits the input $\mathbf{x}$ into two parts: $\mathbf{x} = [\mathbf{x}_a, \mathbf{x}_b]$. The forward transformation is: $\mathbf{y}_a = \mathbf{x}_a, \mathbf{y}_b = \mathbf{x}_b \odot \exp(s(\mathbf{x}_a)) + t(\mathbf{x}_a)$, where $s(\cdot)$ and $t(\cdot)$ are scale and translation functions. The inverse process is: $\mathbf{x}_b = (\mathbf{y}_b - t(\mathbf{y}_a)) \odot \exp(-s(\mathbf{y}_a))$.

The log-determinant of the Jacobian is: $\log\left|\det \frac{\partial \mathbf{y}}{\partial \mathbf{x}}\right| = \sum_i s_i(\mathbf{x}_a)$.

The model stacks multiple coupling layers, where each layer performs the transformation: $\mathbf{z} = f_L \circ f_{L-1} \circ \cdots \circ f_1(\mathbf{x})$. To ensure expressiveness, permutations are applied between layers. The loss function for training is the negative log-likelihood: $\mathcal{L}_{\text{Flow}} = -\log p(\mathbf{z}) - \sum_{i=1}^{L} \log\left|\det \frac{\partial f_i(\mathbf{x})}{\partial \mathbf{x}}\right|$.

### 3.1.3 POLICY GRADIENT METHODS

Policy Gradient (PG) methods (Schulman et al., 2015) are a class of reinforcement learning algorithms that aim to directly optimize a parameterized policy $\pi_\theta(a|s)$ by maximizing the expected cumulative reward: $J(\theta) = \mathbb{E}_{\tau \sim \pi_\theta}\left[\sum_{t=0}^{T} r_t\right]$, where $\tau = (s_0, a_0, s_1, a_1, \ldots, s_T, a_T)$ represents a trajectory sampled from the policy $\pi_\theta$, and $r_t$ is the reward at time step $t$. The policy parameters $\theta$ are updated using the gradient of the expected reward: $\nabla_\theta J(\theta) = \mathbb{E}_{\tau \sim \pi_\theta}\left[\sum_{t=0}^{T} \nabla_\theta \log \pi_\theta(a_t|s_t) r_t\right]$.

## 3.2 NETWORK PIPELINE

As shown in Fig. 2, our pipeline includes three stages: motion learning, audio-motion mapping, and feature refinement. We introduce three stages in detail as follows.

### 3.2.1 MOTION LEARNING

ReenactAnything (Kansy et al., 2024) identifies that motion can be effectively controlled through text embeddings. These embeddings, representing motions such as "standing," "walking," or "running," influence the model via cross-attention inputs, guiding motion synthesis during the denoising process in training videos. To this end, it leverages text embeddings as motion embeddings, which encode rich semantic information, remain free from spatial constraints, and enable injection at multiple stages of the model.

However, ReenactAnything learns a single motion embedding for each reference video, which limits its practicality for large-scale training. Given that our task primarily involves upper-body movements, we design a generalizable motion encoder that learns motion representations across different input videos. Specifically, the motion encoder is initialized using the vision encoding model from the pretrained EVA-CLIP (Sun et al., 2023). To optimize memory efficiency, we further reduce the feature dimensions by incorporating a linear layer.

As illustrated in Stage 1 of Fig. 2, the input video is passed through the motion encoder and a video diffusion model comprising a VAE encoder and a 3D-UNet. The output from the motion encoder is then injected into the cross-attention mechanism of the 3D-UNet. During this process, only the motion encoder is trained using $\mathcal{L}_{\text{LDM}}$.

In this manner, the proposed generalizable motion encoder can generate motion representations for any given input video, enabling broader applicability and scalability.

### 3.2.2 AUDIO-MOTION MAPPING

For modeling the relationship between audio and motion, prior methods (Ginosar et al., 2019; Qian et al., 2021; Corona et al., 2024; Huang et al., 2024; He et al., 2024) typically employ generative models to synthesize motion from audio. However, these methods introduce generation errors, further compounded by inaccuracies in annotated pose data. Additionally, our motion representation is implicit rather than explicit pose data, making generative approaches less suitable for our design. Alternatively, retrieval-based methods (Zhou et al., 2022; Liu et al., 2024a) retrieve ground-truth motions but still depend on annotated poses. Moreover, retrieval-based methods require similarity calculations and ranking during inference, making it computationally inefficient.

In our work, we adopt a dual-tower architecture following retrieval-based methods, as illustrated in Stage 2 of Fig. 2. However, since our motion representation is derived from the motion encoder and is implicit rather than explicit pose information, it is necessary to map the audio information into the motion space of motion encoder during inference. To achieve this, we employ an invertible feature extractor for motion feature extraction. Specifically, we adopt RealNVP (Dinh et al., 2016) as our core architecture, incorporating several linear coupling layers. This design allows the motion representation to be passed through the feature extractor during training, enabling it to learn the audio-motion mapping via the forward process. During inference, the inverse process maps the audio information back into the motion representation, ensuring efficient and consistent audio-to-motion mapping.

In addition, to enhance the expressiveness of the audio-motion joint space and improve its representation power, we model it as a Gaussian mixture, inspired by FlowGMM (Izmailov et al., 2020). Specifically, instead of assuming a single Gaussian distribution, we compute the probability density $\log p(\mathbf{z})$ in $\mathcal{L}_{\text{Flow}}$ using multiple Gaussian components with learnable means $\mu$ and variances $\sigma$. This allows the model to capture more diverse motion patterns and better align with the underlying distribution of real-world audio-motion relationships.

For audio processing, we extract MFCC features and process them with a temporal self-attention layer. Simultaneously, we utilize a pre-trained HuBERT (Hsu et al., 2021) model to extract semantic features. These two feature types are then concatenated and fused through two additional temporal self-attention layers.

---

**Algorithm 1:** Hand Refinement

---

**Input:** Trained video diffusion model VD $(\cdot)$, audio embedding $o$, pre-trained pose estimation
      model DW $(\cdot)$, hand pose confidence threshold $con$, total epochs $N$, learnable
      parameters $\mu, \sigma$.
**Output:** Optimized $\mu, \sigma$.

1   Initialize $\mu = 0$, $\sigma = 1$
2   **for** $i = 1$ to $N$ **do**
3      Sample $z_t \sim \mathcal{N}(\mu, \sigma^2)$; $\hat{z}_t = z_t$
4      **Denoising:** $\hat{z}_0 \leftarrow$ VD $(\hat{z}_t, o)$; **Decoding:** $\hat{x}_0 \leftarrow \hat{z}_0$; **Scoring:** $h \leftarrow$ DW $(\hat{x}_0)$; **Filtering:**
      $\hat{h} \leftarrow h < con$; **Reward:** $r \leftarrow \hat{h} - con$
5      **Loss:** $-\log p(z_t) \cdot r$
6      Update $\mu, \sigma$

7   **return** $\mu, \sigma$

---

Training this stage presents challenges, as $\mathcal{L}_{\text{Flow}}$ often results in a high value. To address this, we initially train the invertible feature extractor using the flow loss $\mathcal{L}_{\text{Flow}}$. Subsequently, the audio encoder and invertible feature extractor are trained jointly using a combination of MSE loss, inter-clip contrastive learning loss, and intra-clip contrastive learning loss. The loss functions are detailed below.

$$\mathcal{L}_{\text{Intra}} = -\frac{1}{N} \sum_{i=1}^{N} \frac{1}{K_i} \sum_{k=1}^{K_i} \log \frac{\exp(\text{sim}(o_{i,k}, m_{f_{i,k}})/\kappa)}{\sum_{l=1}^{K_i} \exp(\text{sim}(o_{i,l}, m_{f_{i,l}})/\kappa)} \tag{1}$$

$$\mathcal{L}_{\text{Inter}} = -\frac{1}{N} \sum_{i=1}^{N} \log \frac{\exp(\text{sim}(o_i, m_{f_i})/\kappa)}{\sum_{j=1}^{N} \exp(\text{sim}(o_i, m_{f_j})/\kappa)} \tag{2}$$

$$\mathcal{L}_{\text{MSE}} = \frac{1}{N} \sum_{i=1}^{N} \|o_i - m_{f_i}\|_2^2 \tag{3}$$

Here, $o$ represents the audio embedding produced by the audio encoder, while $m_f$ denotes the motion embedding obtained through the forward process of the invertible motion feature extractor. $\kappa$ denotes the temperature factor.

The total training loss function is presented below, where $\alpha, \beta, \gamma, \delta$ denote the respective loss weights:

$$\mathcal{L} = \alpha \mathcal{L}_{\text{MSE}} + \beta \mathcal{L}_{\text{Intra}} + \gamma \mathcal{L}_{\text{Inter}} + \delta \mathcal{L}_{\text{Flow}} \tag{4}$$

During inference, given only the audio input, the motion representation used to control the person's movement in the video is derived through the inverse process, as illustrated in Stage 3 of Fig. 2.

### 3.2.3 FEATURE REFINEMENT

Although the motion representation can be effectively learned through Stages 1 and 2, finer-grained details still require improvement. To address this, we introduce a Stage 3, as depicted in Fig. 2. Given an audio input, the audio encoder produces an audio embedding $o$. After completing the training in Stage 2, we assume that the audio embedding $o$ approximates the motion embedding $m_f$. This motion embedding is then passed through the invertible feature extractor to derive the motion representation $m_{inv}$, which is responsible for controlling movements learned during Stage 1. The representation $m_{inv}$ is obtained via inverse process. We train this stage using the loss function $\mathcal{L}_{\text{LDM}}$.

### 3.3 HAND REFINEMENT VIA INITIAL NOISE OPTIMIZATION

Generating realistic hands is challenging due to their diverse appearance features. In human video generation, prior works (Zhang et al., 2024b; Zhou et al., 2024) incorporate pose information during training to enhance quality. However, these methods introduce additional annotation costs and increase model complexity during training. To address this, inspired by prior works (Guo et al., 2024; Chen et al., 2024a) on initial noise optimization, we propose a hand refinement method during sampling based on policy gradient.

Table 1: Quantitative comparison with previous works on four objective metrics.

| Model | FGD ↓ | Div. ↑ | BAS ↑ | FVD ↓ |
|---|---|---|---|---|
| S2G | 3.69 | 180.59 | 0.7280 | 816.03 |
| MYA | 24.24 | 224.14 | 0.7452 | 1823.97 |
| Echo | 22.68 | 233.72 | 0.7427 | 1664.70 |
| **Ours** | **1.11** | **282.89** | **0.7526** | **626.58** |

Specifically, we define $\mu, \sigma$ as the learnable parameters $\theta$, $\mathcal{N}(\mu, \sigma^2)$ as the policy $\pi$, with the sampled latent $\boldsymbol{z}_t$ representing the action $a$, and the input audio embedding $o$ serving as the state $s$. For the reward $r$, we first use the DWPose model (Yang et al., 2023) to estimate the hand pose confidence scores $h$. We then filter and retain only those values $\hat{h}$ that are below the threshold $con$. By defining the reward as $\hat{h} - con$ and maintaining it as a negative value, maximizing the reward brings it closer to zero. The final loss is formulated as:

$$\mathcal{L} = -\log p(\boldsymbol{z}_t) \cdot r, \tag{5}$$

where the negative sign ensures compatibility with gradient descent.

The procedure is detailed in Algorithm 1. After optimization, the parameters $\mu$ and $\sigma$ are refined, allowing $z_t$ to be sampled from optimized distribution $\mathcal{N}(\mu, \sigma^2)$, which enhances hand generation.

# 4 EXPERIMENTS

## 4.1 DATASET

Following previous work (He et al., 2024; Liu et al., 2022a), our dataset is constructed from the PATS dataset (Ginosar et al., 2019; Ahuja et al., 2020a;b), focusing on four individuals: Oliver, Noah, Seth, and Huckabee. The dataset comprises 33 hours of data, with 31.4 hours used for training and the remaining 1.6 hours reserved for testing. Details are provided in the **Appendix**.

## 4.2 EVALUATION METRICS

Following the previous work (He et al., 2024), we assess the quality, diversity, and synchronization between gestures and speech using the following metrics: **Fréchet Gesture Distance(FGD)** (Qian et al., 2021), **Diversity (Div.)** (Liu et al., 2022b), **Beat Alignment Score (BAS)** (Li et al., 2021) and **Fréchet Video Distance (FVD)** (Unterthiner et al., 2018). A detailed introduction to these metrics is provided in the **Appendix**.

## 4.3 COMPARISONS

We compare our method with three open-source approaches, S2G (He et al., 2024), MYA (Huang et al., 2024) and EchoMimicV2 (Meng et al., 2024). While S2G is specifically designed for co-speech gesture video generation, MYA and EchoMimicV2 focus on pose-image-driven synthesis. To adapt MYA and EchoMimicV2 to our setting, we first train DiffSHEG (Chen et al., 2024b) on our dataset to generate pose images as input. To ensure a fair comparison, we fine-tune both models on our dataset. The results are presented in Fig. 3 and Table 1. The quantitative results in Table 1 highlight the advantages of our method. Our model outperforms prior works across four metrics. The superior FGD scores indicate that our approach generates videos that are more aligned with the ground truth. Higher Diversity scores demonstrate the model's ability to produce a broad range of natural gestures. Furthermore, our method achieves the best FVD performance, confirming its capability to synthesize high-quality, realistic gesture videos.

As shown in Fig. 3, S2G, MYA and EchoMimicV2 suffer from noticeable artifacts, particularly blurry and distorted hands. In contrast, our method generates high-quality videos without these issues. Additionally, MYA tends to overfit appearance details from the training data, leading to inconsistencies where the generated frames fail to match the first frame. In addition, the blur in our results naturally arises from motion, whereas in S2G, MYA and EchoMimicV2, it is introduced by their reliance on keypoint movements or pose-image constraints, which limit their ability to capture fine-grained details. More importantly, unlike prior methods, our approach eliminates the need for annotated pose information, relying solely on audio and video during training.

We also conduct a user study, detailed in the **Appendix**.

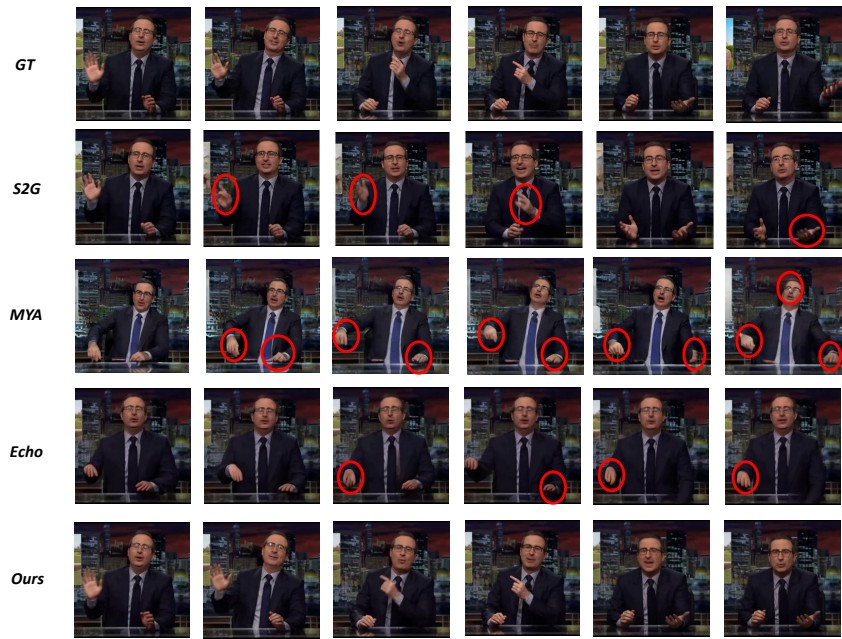

Figure 3: The leftmost image in the GT column represents the first frame. Red circles highlight noticeable artifacts in prior methods. As shown, existing approaches suffer from issues such as blurry hands and distorted fingers. In contrast, our method produces high-quality videos. More importantly, our approach generates videos that are better aligned with the ground truth. Please zoom in for better visibility of details. Video results are shown in **Supplementary Material**.

Table 2: Quantitative ablation study on different stages across four objective metrics. Ours: Stage 3 + hand refinement. *: Upper bound of our method.

| Model | FGD ↓ | Div. ↑ | BAS ↑ | FVD ↓ |
|---|---|---|---|---|
| Stage 1 | 0.92* | 190.82 | 0.7601* | 901.25 |
| Stage 2 | 1.47 | 252.33 | 0.7455 | 932.56 |
| Stage 3 | 1.25 | 275.27 | 0.7503 | 660.73 |
| **Ours** | **1.11** | **282.89** | **0.7526** | **626.58** |

## 4.4 ABLATION STUDIES

**Stage 1: The Effectiveness of Motion Encoder.** Here, we evaluate the effectiveness of our motion encoder. Specifically, we directly test the Stage 1 model by using the test video as input to the motion encoder while keeping the first frame as the appearance feature. As shown in Table 2, the Stage 1 model achieves the best performance on FGD and BAS, demonstrating the superiority of the motion encoder in learning motion representations and its strong generalization ability. However, the lower FVD performance indicates that fine-grained details still need to be learned, particularly in human-centric scenarios, as further validated in Fig. 4. It is important to note that the FGD and BAS scores at this stage represent the upper bound of our final model, as the test video is directly used for verification. Additionally, Stage 2 introduces unavoidable errors due to the imperfect audio-to-motion mapping, inevitably leading to a decline in these metrics in subsequent stages.

**Stage 2: Without Invertible Feature Extractor**. During Stage 2 evaluation, we apply a similar inverse process as in Stage 3, but use the untrained diffusion model from Stage 1. To verify the necessity of feature learning for motion representation, we remove the invertible feature extractor in Stage 2. As shown in Table 3, the performance across all four metrics drops significantly compared to Stage 2 with the feature extractor, demonstrating the effectiveness of our invertible feature extractor. The visual results, shown in Fig. 4, further highlight this issue, where the generated gestures fail to align with the ground truth and exhibit noticeably degraded visual quality.

**Stage 3: The Effectiveness of Feature Refinement.** As shown in Fig. 4 and Table 2, Stage 3 plays a crucial role in enhancing visual quality, as reflected in the FVD metric. Additionally, since motion

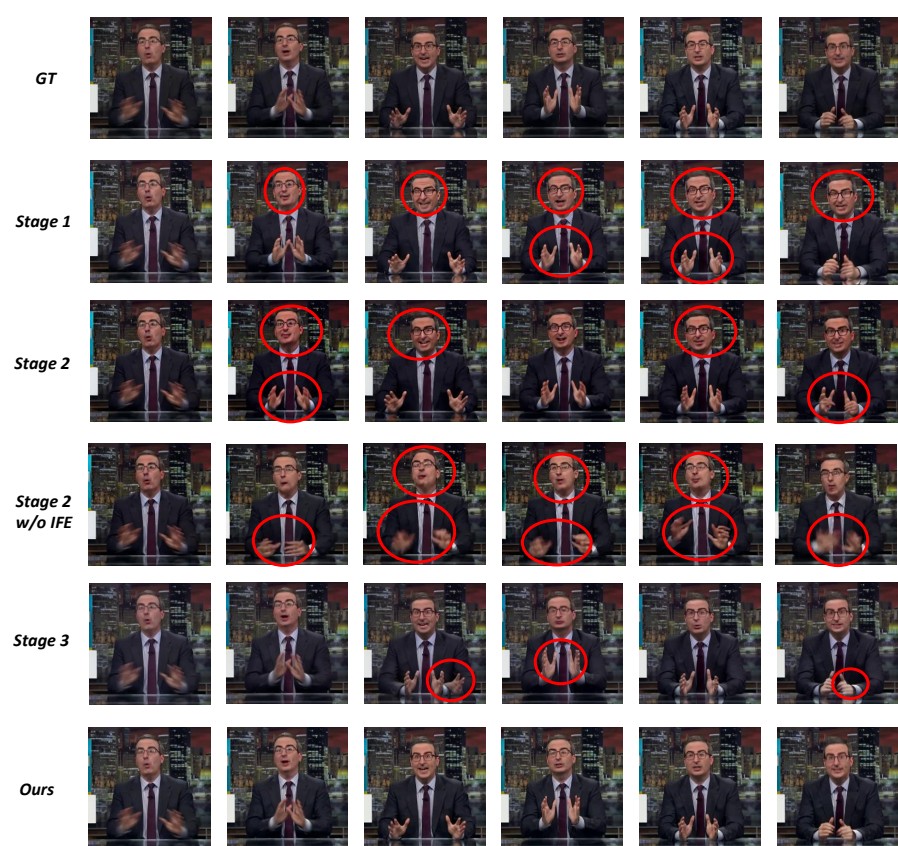

Figure 4: As shown, Stage 1 and Stage 2 produce well-aligned gestures but lack fine-grained details. Removing the invertible feature extractor in Stage 2 results in misaligned gestures and degraded visual quality. While Stage 3 enhances visual quality, it still struggles with hand generation. By applying our hand refinement method, the final model generates high-quality videos. Please zoom in for better visibility of details. Video results are shown in **Supplementary Material**.

Table 3: Results of the model without the invertible feature extractor in Stage 2 across four objective metrics.

| Model | FGD ↓ | Div. ↑ | BAS ↑ | FVD ↓ |
|---|---|---|---|---|
| w/o IFE | 45.73 | 175.08 | 0.7289 | 2156.39 |
| Stage 2 | 1.47 | 252.33 | 0.7455 | 932.56 |

information can also be captured through temporal attention in the video diffusion model, Stage 3 achieves slight improvements over Stage 2 in FGD, Div., and BAS.

**t-SNE Projection of Audio-Motion Embeddings.** As shown in Fig. 5, we visualize the t-SNE projection of the learned motion embeddings, colored by motion intensity. The samples naturally organize into three distinct clusters: Silent, Calm Gesture, and Intense Motion. Notably, the Silent cluster is located in close proximity to the Calm Gesture cluster, reflecting their semantic similarity in terms of low-to-moderate motion energy. In contrast, the Intense Motion cluster is clearly separated from the other two, occupying a distant region in the embedding space. This topological arrangement demonstrates that the learned embeddings capture meaningful motion dynamics: the embedding distances correlate with the magnitude of motion intensity rather than static appearance cues, effectively distinguishing between subtle gestures and high-energy movements.

Additional ablation studies on **different loss functions** and **hand refinement** are shown in the **Appendix**.

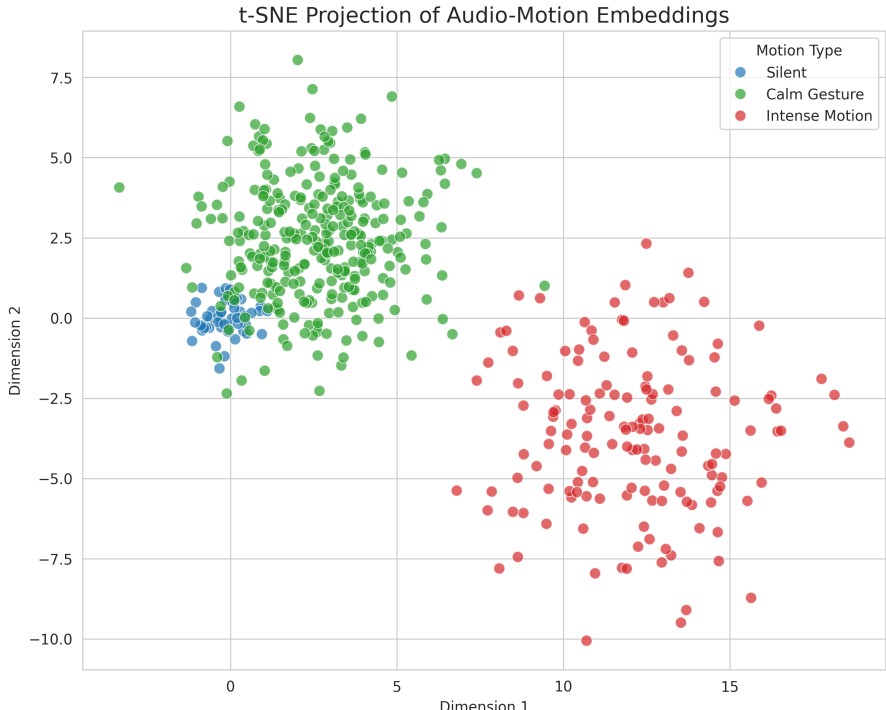

Figure 5: t-SNE Projection of Audio-Motion Embeddings.

## 5 LIMITATIONS

Since our method follows the implicit motion extraction introduced in Reenact Anything (RA) (Kansy et al., 2024), our framework naturally inherits both its strengths and limitations. Regarding the potential appearance–motion entanglement, RA reports that pre-trained image-to-video models primarily derive appearance information from the latent image input, while the text/image embeddings injected through cross-attention mainly control motion. The same mechanism applies in our setting, where the output of our motion encoder is injected into the diffusion model via cross-attention, thereby mainly guiding motion.

As for failure cases, the model performs less reliably under large camera/body movements. This is partly due to the limitation that the motion extraction relies on a pre-trained video generation model, which inherently struggles with large movements. Additionally, our current training set is predominantly composed of front-facing videos and does not include sufficient examples of large body movements or diverse camera dynamics.

## 6 CONCLUSION

In this work, we introduce a weakly supervised motion learning framework for co-speech gesture video generation that eliminates the need for pose supervision while achieving state-of-the-art performance. Our approach leverages a motion encoder to learn a generalizable motion representation directly from video, a dual-tower architecture to align audio with motion using an invertible feature extractor, and a video diffusion model to refine fine-grained details. Additionally, we propose a novel hand refinement strategy based on initial noise optimization, improving hand synthesis. Extensive experiments on our collected dataset demonstrate the effectiveness of our framework, achieving superior performance over existing methods, especially on audio-to-gesture alignment and hand generation. By eliminating the reliance on pose annotations and introducing a more efficient motion-learning paradigm, our work establishes a new state-of-the-art in co-speech gesture video generation.

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

Table 4: Comparisons of different loss functions in Stage 2 on four objective metrics.

| Inter | Intra | MSE | FGD $\downarrow$ | Div. $\uparrow$ | BAS $\uparrow$ | FVD $\downarrow$ |
|-------|-------|-----|------|------|------|------|
| ✓ | | | 5.53 | 202.36 | 0.7339 | 1223.06 |
| ✓ | ✓ | | 3.95 | 221.25 | 0.7392 | 1105.89 |
| ✓ | ✓ | ✓ | 1.47 | 252.33 | 0.7455 | 932.56 |

Table 5: Results of the mean hand pose confidence score without or with hand refinement. Ours: Stage 3 + hand refinement.

| Model | Mean Hand Pose Confidence $\uparrow$ |
|-------|--------------------------------------|
| Stage 3 | 88.73% |
| Ours | 95.45% |

## A  ADDITIONAL ABLATION STUDIES

**Stage 2: Different Loss Functions.** Here, we provide quantitative results on the impact of different loss functions in Stage 2 for aligning audio and motion. As shown in Table 4, the intra-clip contrastive learning loss and MSE loss are important for accurate audio-to-motion mapping.

**Hand Refinement.** As shown in Fig. 4 and Table 2, applying hand refinement during sampling significantly enhances hand generation quality, particularly in visual fidelity. Since body metrics are insufficient for assessing hand quality, we introduce *Mean Hand Pose Confidence* to evaluate the realism of hand poses. This metric measures the anatomical naturalness of hands by averaging the confidence scores predicted by DWPose (Yang et al., 2023) across all detected hand keypoints. As shown in Table 5, incorporating hand refinement during sampling results in a substantial improvement in hand synthesis. Moreover, we observe that as hand quality improves, overall visual quality—including facial detail—also benefits. This suggests a strong correlation between hand realism and the overall perceptual quality of generated videos.

## B  IMPLEMENTATION DETAILS

We use the Adam optimizer for all three training stages as well as for hand refinement during sampling. The input image resolution is set to $512 \times 512$. To expose the model to more context, training videos are sampled at 7 FPS, allowing the video diffusion to process approximately 2 seconds of video with 16 frames. The inference timestep is set to 50. The classifier-free guidance (CFG) scale is set to 3.5. For long video generation, we first generate a short clip and use its final frame as the first frame for the next clip generation. Specific setting for each stage is listed below.

**Stage 1**. The motion encoder is initialized with the visual encoder from a pretrained CLIP model [1]. To reduce memory consumption, the feature dimension is reduced to $5 \times 1024$ via a linear layer. Since video motion is largely determined during noisy diffusion steps, we shift the noise schedule toward higher noise values using a noise offset of 0.1 to accelerate optimization. This stage is trained for 50k steps with a learning rate of $1e^{-4}$, using 4 A100 GPUs for 1 day with a batch size of 1. To improve efficiency, 16 fixed frames are sampled from each video clip.

**Stage 2**. We first train the invertible feature extractor with a batch size of 1 and a learning rate of $1e^{-4}$ for 200k steps. Next, we train the audio-motion mapping with a batch size of 16 and a learning rate of $1e^{-3}$ for 20k steps. The entire second stage is trained on 8 RTX 8000 GPUs for 1 day. The number of Gaussian components is set to 5. For the invertible feature extractor, the input dimension is 1024, the coupling layer dimension is 512, and the model has a depth of 12 layers. Since the original audio is sampled at 30 FPS while the videos are at 7 FPS, an additional Conv1D-based downsampling layer is applied during audio encoding. The loss weights $\alpha, \beta, \gamma, \delta$ are empirically set to $5, 3, 1, 0.01$, respectively. During training, 16 frames are randomly selected per clip, with the starting index chosen from a list with a stride of 14.

---

[1]https://huggingface.co/QuanSun/EVA-CLIP/blob/main/EVA02_CLIP_L_psz14_s4B.pt

**Stage 3**. The model is trained with a batch size of 1 and a learning rate of $1e^{-5}$ for 150k steps, using 4 A100 GPUs for 3 days. The downsampling layer from Stage 2 is reused, and training clips are sampled with 16 frames, selecting the starting index from a list with a stride of 14.

**Hand Refinement**. The hand pose confidence threshold $con$ is set to $95\%$. The total number of epochs $N$ is set to 30, with a learning rate of $1e^{-2}$.

## C    METRICS

### C.1    DEFINITIONS

- **Fréchet Gesture Distance (FGD)** (Qian et al., 2021): Measures the distributional discrepancy between real and synthesized gestures in the feature space.

- **Diversity (Div.)** (Liu et al., 2022b): Computes the average feature distance among generated gestures, indicating their variability.

- **Beat Alignment Score (BAS)** (Li et al., 2021): Evaluates the temporal coherence between speech and gestures by calculating the mean distance between speech beats and gesture beats.

- **Fréchet Video Distance (FVD)** (Unterthiner et al., 2018): Assesses the overall fidelity of gesture videos using the I3D (Wang et al., 2019) classifier trained on Kinetics-400 (Kay et al., 2017).

### C.2    IMPLEMENTATION DETAILS

To assess the quality of the generated videos, we first extract skeleton keypoints using the DWPose framework (Yang et al., 2023). This process retains 12 upper-body keypoints and 21 keypoints per hand, totaling 54 keypoints. FGD and Diversity are computed using an autoencoder trained on skeleton keypoints from our dataset, following the approach outlined in (Qian et al., 2021). Additional details on autoencoder training and FGD computation can be found in their GitHub repository[2]. For Diversity, we adopt the methodology from (Zhu et al., 2023), with implementation details provided in their GitHub repository[3].

Furthermore, BAS is calculated following the method in (Li et al., 2021)[4]. For FVD, we utilize the I3D classifier (Wang et al., 2019), pre-trained on the Kinetics-400 dataset (Kay et al., 2017). Additional implementation details for this metric can be accessed via the corresponding GitHub repository[5].

## D    DATA PROCESSING

Identity labels are used to automatically download videos from YouTube[6], followed by filtering and processing. To ensure high-quality co-speech gesture videos, we apply four filtering principles: (1) Scene Consistency – Videos are segmented using SceneDetect[7] to separate clips with different scenes. (2) Single-Speaker Constraint – TalkNet (Tao et al., 2021) is used for speaker diarization, filtering out multi-person videos. (3) Face Visibility – MediaPipe (Lugaresi et al., 2019) detects faces, and clips with low detection confidence (e.g., side views) are discarded. (4) Minimum Duration – Only clips longer than 3 seconds are retained to ensure they contain meaningful gestures. After filtering, the videos are resampled at 7 FPS. Frames are cropped using square bounding boxes centered on the speaker and resized to $512 \times 512$. This results in a dataset of 33 hours, with 31.4 hours allocated for training and the remaining 1.6 hours for testing.

---

[2]https://github.com/ShenhanQian/SpeechDrivesTemplates

[3]https://github.com/Advocate99/DiffGesture/tree/main

[4]https://github.com/google-research/mint

[5]https://github.com/JunyaoHu/common_metrics_on_video_quality

[6]https://github.com/yt-dlp/yt-dlp

[7]https://github.com/Breakthrough/PySceneDetect

Table 6: Results of the model without motion information across four objective metrics.

| Model | FGD ↓ | Div. ↑ | BAS ↑ | FVD ↓ |
|---|---|---|---|---|
| w/o Motion | 21.95 | 189.02 | 0.7400 | 2406.91 |
| Stage 3 | 1.25 | 275.27 | 0.7503 | 660.73 |

Table 7: Quantitative comparison with previous works on four subjective metrics. Bold text indicates the best performance.

| Model | Preservation ↑ | Quality ↑ | Consistency ↑ | Synchronization ↑ |
|---|---|---|---|---|
| S2G | 2.87 | 2.76 | 2.92 | 2.81 |
| MYA | 1.53 | 1.57 | 1.61 | 1.56 |
| EchoMimicV2 | 2.23 | 2.27 | 2.32 | 2.28 |
| **Ours** | **3.62** | **3.73** | **3.53** | **3.72** |

## E  ADDITIONAL RELATED WORK: MOTION TRANSFER

Fine-tuning approaches, such as VMC (Jeong et al., 2024), combine fine-tuning with inversion through adaptive temporal layer adjustments, achieving superior motion transfer results while maintaining temporal consistency. Similarly, Tune-a-Video (Wu et al., 2023a) adapts text-to-image models for motion transfer by adding spatio-temporal attention layers, training only motion-specific components. MotionDirector (Zhao et al., 2024) innovates with a dual-path LoRA architecture to separate motion and appearance learning, enabling precise control over temporal dynamics. DreamVideo (Wei et al., 2024) and Customize-A-Video (Ren et al., 2024) further decouple spatial and temporal information through distinct branches for appearance and motion learning, although they still face challenges with appearance-motion coupling. Motion Inversion (Wang et al., 2024b) learns motion embeddings using temporal attention layers trained directly on reference videos, providing precise temporal control while maintaining visual quality.

Other approaches extract motion representations at inference, such as DMT (Yatim et al., 2024) and MOFT (Xiao et al., 2024), suitable for cross-architecture applications. DMT introduces a space-time feature loss that utilizes DDIM inversion and UNet activations, while MOFT discovers motion channels in diffusion features.

## F  WITHOUT MOTION INFORMATION

To verify the necessity of motion information, we directly train Stage 3 with a trainable audio encoder while removing the invertible feature extractor. The generated videos exhibit severe artifacts such as distorted hands, extra fingers, and hands appearing detached from the body (see video results). These issues suggest that relying solely on audio leads to a weak correlation between audio and motion during training, ultimately degrading generalization performance during testing. Furthermore, this setup results in lower scores across objective metrics, as shown in Table 6, particularly in FVD, indicating that relying exclusively on weak audio signals leads to poorer overall visual quality.

## G  USER STUDY

To further assess the visual quality of our approach, we conduct a user study comparing gesture videos produced by different methods. We randomly select 30 generated videos from our test set for each method and invite 20 participants to evaluate and rank them. The evaluation is based on the following four criteria:

- **Speech-Gesture Synchronization**: Measures how well the generated gestures align with the speech, ensuring accurate motion timing.

Table 8: Statistical Comparison of Methods.

| Metric | Reference | Compared | Mean Difference | T Statistic | P Value | Significant |
|---|---|---|---|---|---|---|
| Preservation | Ours | S2G | 0.7489 | 13.0230 | $3.142 \times 10^{-34}$ | TRUE |
| Preservation | Ours | MYA | 2.0923 | 41.7407 | $3.475 \times 10^{-178}$ | TRUE |
| Preservation | Ours | EchoMimicV2 | 1.3876 | 28.0123 | $3.289 \times 10^{-61}$ | TRUE |
| Quality | Ours | S2G | 0.9632 | 17.8820 | $9.321 \times 10^{-55}$ | TRUE |
| Quality | Ours | MYA | 2.1490 | 39.9311 | $1.592 \times 10^{-169}$ | TRUE |
| Quality | Ours | EchoMimicV2 | 1.4587 | 29.1234 | $3.678 \times 10^{-63}$ | TRUE |
| Consistency | Ours | S2G | 0.6078 | 6.5864 | $6.089 \times 10^{-11}$ | TRUE |
| Consistency | Ours | MYA | 1.9145 | 28.6291 | $1.298 \times 10^{-112}$ | TRUE |
| Consistency | Ours | EchoMimicV2 | 1.2034 | 15.7890 | $3.145 \times 10^{-36}$ | TRUE |
| Synchronization | Ours | S2G | 0.9091 | 16.5028 | $2.198 \times 10^{-52}$ | TRUE |
| Synchronization | Ours | MYA | 2.1521 | 40.7988 | $4.219 \times 10^{-176}$ | TRUE |
| Synchronization | Ours | EchoMimicV2 | 1.4390 | 29.5123 | $3.512 \times 10^{-65}$ | TRUE |

Table 9: Mean Scores and Standard Deviations for Each Method.

| Video | Metric | Mean Score | Std Dev |
|---|---|---|---|
| Ours | Preservation | 3.6234 | 0.5978 |
| Ours | Visual Quality | 3.7345 | 0.6123 |
| Ours | Consistency | 3.5276 | 0.6890 |
| Ours | Synchronization | 3.7190 | 0.5876 |
| S2G | Preservation | 2.8732 | 0.4890 |
| S2G | Visual Quality | 2.7643 | 0.4789 |
| S2G | Consistency | 2.9167 | 0.5012 |
| S2G | Synchronization | 2.8090 | 0.4921 |
| MYA | Preservation | 1.5334 | 0.4123 |
| MYA | Visual Quality | 1.5734 | 0.4390 |
| MYA | Consistency | 1.6078 | 0.4456 |
| MYA | Synchronization | 1.5590 | 0.4289 |
| EchoMimicV2 | Preservation | 2.2334 | 0.4123 |
| EchoMimicV2 | Visual Quality | 2.2734 | 0.4190 |
| EchoMimicV2 | Consistency | 2.3123 | 0.4234 |
| EchoMimicV2 | Synchronization | 2.2790 | 0.4167 |

- **Identity Preservation**: Evaluates how faithfully the generated video preserves the subject's defining characteristics and appearance.

- **Temporal Consistency**: Assesses the continuity and fluidity of motion across consecutive frames, ensuring natural movement transitions.

- **Visual Quality**: Examines the overall image quality, with higher ratings indicating fewer distortions, blurring, or noise artifacts.

Participants rank the videos, with rank 1 representing the best quality. To enable fair comparisons with prior works, rankings are converted into weighted scores: rank 1 receives 4 points, rank 4 is assigned 1 point, and so forth. A higher cumulative score indicates stronger overall performance.

The user study findings are summarized in Table 7. As evidenced in the table, our method consistently surpasses prior approaches across all evaluation criteria, highlighting its effectiveness in generating high-quality gesture videos with enhanced motion accuracy and visual fidelity.

The ABX test results, presented in Tables 8 and 9, reveal statistically significant differences across all evaluated metrics when comparing the methods, with all p-values below 0.05. Our approach consistently surpasses all the S2G, MYA and EchoMimicV2 methods across every quality measure, achieving higher mean scores ranging from 3.53 to 3.73, compared to S2G's 2.76 to 2.92, MYA's

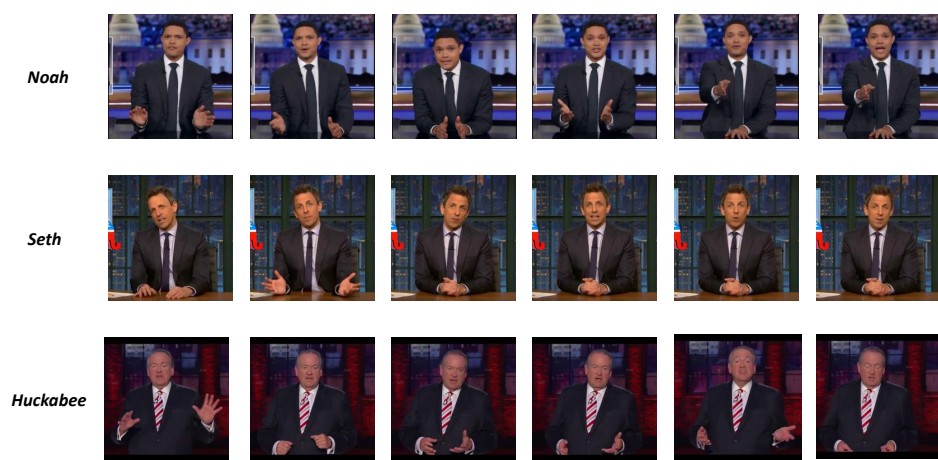

Figure 6: Visual results of the other three identities.

1.53 to 1.61, and EchoMimicV2's 2.23 to 2.31. The most substantial performance gaps appear between our method and MYA, where mean differences exceed 2 points for the Identity Preservation, Visual Quality and Speech-Gesture Synchronization metrics. Although S2G outperforms MYA and EchoMimicV2, it still falls significantly short of our method, particularly in Visual Quality, where the mean difference reaches 0.97. These findings provide strong evidence of our method's superior performance in video quality, further supported by high t-statistics and extremely low p-values across all comparisons.

## H    EFFICIENCY COMPARISON

We compare the inference time and memory usage of our method against existing approaches. For generating a 1-second video at 30 FPS with a resolution of $512 \times 512$ on an A100 GPU, S2G requires 4.6 seconds and 3GB of memory, MYA takes 40 seconds and 46GB, while our method completes in 16.6 seconds using 27GB. Compared to the diffusion-based MYA, our method is more efficient, achieving faster inference speed while requiring less memory.

It is important to note that our model is trained and tested at 7 FPS. However, for a fair comparison with prior works, we generate 30 frames in this evaluation.

## I    MORE RESULTS ON OTHER IDENTITIES

In the main text, we mainly show the results of Oliver identity, here we attach the visual results of the other three identities (Fig. 6).

## J    FUTURE WORK

While our method achieves strong performance in co-speech gesture video generation, there remains room for further advancements. Below, we outline key areas for future exploration.

**Video Enhancement**. To provide our model with richer contextual information, we currently use 7 FPS videos. Future work can explore integrating existing video enhancement techniques to improve temporal consistency and overall visual quality.

**Extending Hand Refinement to Other Tasks**. Our hand refinement approach, based on initial noise optimization via reinforcement learning, has broader applicability beyond co-speech gesture video generation. Investigating its potential in tasks such as pose-driven human video generation could further expand its impact.

**Larger Dataset and Model**. Expanding the dataset and evaluating our method on a larger scale will be a key focus for future research. Additionally, to remain competitive with other foundation model-based approaches, further architectural refinements are necessary to accommodate larger models.

Future improvements should enhance lip-sync performance, enable zero-shot generalization, and extend to diverse scenarios beyond front-facing video generation.

