# OpenReview forum: "Weakly Supervised Motion Learning for Co-speech Gesture Video Generation"
_ICLR.cc/2026/Conference — Submitted to ICLR 2026_

### Official Review · Reviewer_ptuf · 2025-10-15

**Soundness:** 2
**Presentation:** 1
**Contribution:** 2
**Rating:** 4
**Confidence:** 4

**Summary:**

This paper proposes a weakly supervised framework for co-speech gesture video generation that eliminates the need for pose annotations by learning motion directly from raw video and audio. The method consists of three stages: (1) a motion encoder that learns a generalizable motion representation from video using latent diffusion; (2) a dual-tower audio–motion mapping module with an invertible feature extractor that aligns audio embeddings to motion space without retrieval; and (3) a video diffusion model that refines fine-grained visual details. Additionally, a policy-gradient-based hand refinement optimizes the diffusion noise distribution to improve hand realism during sampling. Experiments on a 33-hour dataset demonstrate state-of-the-art performance across motion fidelity, diversity, and realism metrics, outperforming prior works such as S2G, MYA, and EchoMimicV2—all without requiring pose supervision.

**Strengths:**

The video generation quality for the pixel-synthesis quality in general looks fine. The hands are not blurry, which is good. Not very sinigificant deformation issues.

The design of the hand refinement and other types of refinement also looks good to me. It is a good design for the pixel-quality enhancement.

**Weaknesses:**

Overall, i guess the major problem would be that the authors have claimed too many things in the paper but do not have very solid experiments to validate designs, making the paper more like an integration of tricks instead of in-dpeth discussion and analysis of the proposed solutions for co-speech gesture generation model.

1) Clarity of the “motion learning” module

The current description of the motion-learning block is under-specified. From the text it is unclear how this module is trained or even what objective signals drive learning. Is it contrastive, retrieval-margin (InfoNCE / triplet), reconstruction, diffusion, or hybrid? Please specify the exact loss.

2) Is it really a good motion representation
For Feature space, the author claimed they first leverage a EVA-CLIP to extract motion feature, so it is implicit. However, it is uncertain why this motion representation is really good or not. What if utilize another encoder? How the temporal dynamics are learned from this motion representation? How does it differ from the optical-flow-based methods (MRAA, TPS, or even just explicit 2d poses, etc)? A high-level discussion of this is really important. It would be better if the authors can provide the analysis and comparison of their proposed representation with the existing ones. I can take an example experiment, for TPS transformation as the motion representation, it can be easily integrated into the authors' framework, to achieve this goal, the authors can keep the deformation network and feature extraction network from TPS, replacing the decoder of the original TPS paper with the current Stable-Diffusion as the backbone. This will let the readers to know a solid comparison of motion representations and see if this is good design.


3) Positioning vs. closely related work

The manuscript mentions a “retrieval-based setting” but does not clearly position itself against recent audio→motion literature. Please add a comparison table and narrative against:

TANGO (ICLR 2025): Audio-conditioned gesture generation with temporal alignment constraints (if applicable). What differs in your alignment strategy, representation choice, and objective?

Contextual Gesture (ACM MM 2025): Uses explicit keypoints and a masking/infilling objective. Are you using a different masking schedule, context modeling, or a retrieval head instead of generation? Why is your choice better for downstream video synthesis?

Hierarchical Cross-Modal Association (CVPR 2022): Introduces a hierarchical association between modalities. Do you model hierarchy (phoneme→word→phrase) or stay flat? If flat, discuss trade-offs.


3) Why not a generative motion mapper?

Given the two-stage design (first generate motion, then video), a purely retrieval-trained motion space leaves open the question of temporal fidelity and sample quality at inference.

Please compare against audio-conditioned generative baselines:

ANGIE (NeurIPS 2022): Audio→MRAA latent generative model.

S2G-Diffusion (CVPR 2024): Audio→TPS diffusion.

Contextual Gesture (ACM MM 2025): Audio→keypoint masking/generation.

For the motion mapping, what if the authors train these types of generative models from audio to latent motion?

4) How can we trust motion quality?

As long as you have the motion representations, the motion quality is then being able to evaluate to let readers to see if quantitative evidence that the intermediate motion is good and useful.

I would recommended evaluations like:

Motion FID / KID in the motion-feature space (compute features via a frozen motion-recognition backbone or your own motion encoder).

Fréchet Gesture Distance (FGD) or DTW-based alignment between generated and reference motion.

Diversity & coverage: average pairwise diversity, nearest-neighbor coverage over the test set.

Sync metrics: Audio–motion beat/peak alignment score; landmark velocity–onset correlation.

**Questions:**

During the Rebuttal period, would the authors provide the long-sequence gesture generation videos? The current ones are all short videos, very hard to evaluate if there would be repeated gesture pattern issues, and long dependency issues now.

What is the sampling rate for the video for training and generation, it looks like it is less than 30 frames per second, and look a little bit unnatural.

The results for EchomimicV2 look strange. The raw EchomimicV2 is not supposed to look like that.

Please see the weakness as above.

---

> ### Author Response · Authors · 2025-11-24
>
> Thank you for acknowledging our generation quality and method design.
>
> **W1: Motion Learning Loss**
>
> Sorry for the confusion. In Stage 1, the diffusion model is frozen and only the motion encoder is trained using the diffusion loss, as illustrated in Figure 2.
>
> **W2: Motion representation**
>
> Thank you for the question. Our motion extraction follows the Reenact Anything (RA) method, which uses CLIP embeddings for initialization; accordingly, we initialize our motion encoder with EVA-CLIP. Temporal dynamics are learned because, as reported by RA, pre-trained image-to-video models primarily extract appearance from the latent image input, while text/image embeddings injected via cross-attention control motion.
>
> Compared to optical-flow or pose-based methods (e.g., TPS, MRAA, explicit 2D poses), our motion representation is implicit, requiring no additional annotations or explicit motion computation. This reduces computational overhead and avoids potential errors from rigid supervision. By removing unnecessary strong visual constraints, our method gains flexibility while still capturing motion from the original video.
>
> Notably, the baseline S2G already uses TPS for motion representation, yet our approach achieves better results, demonstrating the effectiveness of our implicit motion design.
>
> Reenact Anything: Semantic Video Motion Transfer Using Motion-Textual Inversion. Kansy, Manuel, et al.  SIGGRAPH 2025.
>
> **W3: Comparisons with related work**
>
> Thank you for the suggestions. Our task focuses on learning and reproducing the unique gesture style of a specific individual, which makes retrieval-based methods more suitable, as they provide a more accurate mapping and less diversity than generative-based methods. Below, we discuss the comparisons with the suggested methods:
>
> TANGO (ICLR 2025): While TANGO also uses contrastive learning, it relies on explicit 3D motion, which can introduce annotation errors. Moreover, TANGO generates the full motion sequence via interpolation, which is different from our approach. TANGO is not fully open-sourced, we are unable to perform experimental comparisons.
>
> Contextual Gesture (ACM MM 2025): Contextual Gesture uses contrastive learning on explicit keypoints with masking/infilling objectives. We do not employ any masking schedule or context modeling. Since it is not open-sourced, we are unable to perform direct comparisons.
>
> Hierarchical Cross-Modal Association (CVPR 2022): This work models hierarchical associations for gesture generation only, not full gesture video generation. Our task differs significantly, making a direct comparison difficult.
>
> S2G-Diffusion (CVPR 2024) and ANGIE (NeurIPS 2022): We have already compared S2G, which in turn outperforms ANGIE. This demonstrates that our approach is effective compared to generative audio-to-motion baselines.
>
> Overall, our retrieval-based design better captures personalized gesture styles for video synthesis, avoids annotation errors, and ensures temporal fidelity, which is crucial for reproducing individual motion patterns.
>
> **W4: Motion Quality**
>
> Thank you for the suggestions. We already evaluate motion quality in Tables 2 and 3. In Table 2, Stage 2 shows effective audio–motion mapping, compared to Stage 1 in terms of FGD, diversity, and BAS. It is important to note that Stage 1 scores represent an upper bound, as the test video is directly used for verification. Table 3 further demonstrates the effectiveness of our invertible feature extractor, highlighting the quality of the learned motion representations.
>
> **Q1: Long video generation**
>
> Thank you for the suggestion. We will include these results in the revised supplementary material.
>
> **Q2: Sampling rate**
>
> Thank you for the question. As mentioned in the appendix, due to computational limitations and to allow the model to see more context, we use a 7 FPS sampling rate. In future work, we plan to explore better 3D VAE-based models or video frame interpolation to further improve visual quality.
>
> **Q3: EchoMimicV2**
>
> EchoMimicV2 and other zero-shot methods target commercial digital human scenarios and do not capture diverse gesture patterns; they primarily focus on lip-sync and limited gesture motion. As evidenced by their official demonstrations, these methods often restrict generation to a limited subset of pre-selected poses to maintain stability. In contrast, our task aims to learn personalized gesture styles, which often involve more diverse motions and fall outside the dataset distribution of EchoMimicV2. This explains the comparatively lower performance of such zero-shot methods on our task.

---

### Official Review · Reviewer_fVo4 · 2025-10-31

**Soundness:** 2
**Presentation:** 2
**Contribution:** 2
**Rating:** 4
**Confidence:** 4

**Summary:**

This paper's motivation is training co-speech gesture video generation model without pose annotation.
1. The task is a audio-text-image to video model (ATI2V) for talking avatar cases.  The detail of the backbone is unclear but it's a latent diffusion model (LDM) based on U-Net. The dataset is open-source 33 hours fix-camera talking dataset (S2G).
2. The idea is first learn a implict motion representation/encoder for video, then train a auido-to-motion model, and finally train the audio-to-motion-to-video model.
    -   motion represenation learning: freeze all parameters of the LDM including VAE, finetune the CLIP image encoder (authors called motion encoder).
    -   audio to motion: freeze finetuned motion encoder. train a audio encoder and a motion mapper with MSE + Contrastive + Flow loss.
    -   a2m2v: only finetune the LDM part.
3. The evaluation have both objective scores and 10+ subjective videos.

**Strengths:**

1. The design of each module has good reference, for example, mulit-mode flow, invertiable motion encoder etc.
2. The results outperforms current method in both objective metrics and subjective videos.
3. The paper is organized and could follow.
4. The idea of finetuning CLIP to reconstuct video is interesting and has insight.

**Weaknesses:**

1. The motivation of pose annotation-free maybe not strong in current ATI2V model.
    -    Due to the good pretrained base model, from the early of 2025, new ATI2V models (Hunyuan video avatar, etc) dont require pose to train the speech-to-gesture model, end2end modeling with a simple audio adapter has been demonstrate enough for basic audio-gesture alignment, such as beat, lip-sync, emotion alginment. could the author explain (in theroy is also fine) why the proposed method is better than such end2end modeling?
2. The performance of finetuning CLIP to reconstruct video is unclear.
    -    Could CLIP or finetuned CLIP model accurate spatial positions of the motion? or just semantic? I just confused if it could why we need ControlNet.
    -    If just semantic, what level it could archieve? could it understand the details of finger motion for example. The abality of its representation is better than pose or even worse?
3. The purpose for some loss terms for audio-to-motion part is unclear.
    -    what is loss flow
    -    why using MSE and Contrastive loss together? seems a combination for single-mode and multi-mode loss without reasons.
4. Ablation study for stage 3.
    -    This is similar to weakness 1. Consider the audio encoder is freezed here, how about replace it to a freezed hubert or wav2vec2, will it also work?
    -    The finetuning of LDM here is finetuning all parameters or only audio related parameters?

**Questions:**

Overall this is an organized paper and I could follow, it has both objective and subjective evaluations. My major concern is the high level motivation. since the authors design a relativly complex three stage system, it requires strong evidence in theroy or results it works better than simple end2end modeling. I'm also interested in the discussion in the CLIP finetuning part. and the other questions are optional.

---

> ### Author Response · Authors · 2025-11-24
>
> Thank you for acknowledging our module design, experimental performance, presentation, and idea.
>
> **W1: Comparison to zero-shot methods**
>
> Thank you for the question. The methods mentioned (e.g., Hunyuan Video Avatar) focus on zero-shot audio-driven video generation, which fundamentally differs from co-speech gesture video generation. Zero-shot methods typically rely on datasets with many identities but only a few seconds per identity (e.g., Vlogger: 2.2k hours / 800k identities, ~9.9s per identity), aiming mainly for lip synchronization with limited hand motion diversity.
>
> In contrast, our task focuses on learning and reproducing the unique gesture style of a specific individual, which requires hours of video per identity. Zero-shot methods cannot capture personalized gestural nuances, as they are designed for generic motion across many identities via large-scale training. By targeting fewer identities with more videos per identity, our method enables annotation-free learning of personalized co-speech gestures, which, to the best of our knowledge, is the first approach to achieve this in co-speech gesture video generation.
>
> **W2: Discussion of CLIP fine-tuning**
>
> Thank you for the question. Our motion extraction follows the Reenact Anything (RA) method, which demonstrates strong spatial understanding, such as automatic spatial alignment and multi-object movement, for tasks like full-body/face reenactment and motion transfer, as illustrated in its Figure 1. Our task primarily involves front-facing, upper-body–centered videos, which closely aligns with RA’s reenactment setting. By replacing the motion embedding in RA with our learnable motion encoder, our model shows similar spatial understanding ability. That said, ControlNet is still necessary for tasks where the original motion video is not available, since our design relies on extracting motion directly from the input video.
>
> Compared to pose-based methods, our primary motivation is to eliminate annotation cost and potential errors. By removing pose supervision, our model gains flexibility and does not require rigid pose alignment, while still producing high-quality motion generation.
>
> Reenact Anything: Semantic Video Motion Transfer Using Motion-Textual Inversion. Kansy, Manuel, et al.  SIGGRAPH 2025.
>
> **W3: Loss terms in stage 2**
>
> Thank you for the question, and we apologize for any confusion. The flow loss is described in Section 3.1.2 and is used to train the invertible feature extractor. The combination of MSE and contrastive learning is inspired by EmotiveTalk. Table 4 in the appendix demonstrates the effectiveness of different loss term combinations.
>
> EmotiveTalk: Expressive Talking Head Generation through Audio Information Decoupling and Emotional Video Diffusion, Haotian Wang et al. CVPR 2025
>
>
> **W4: Ablation study for stage 3**
>
> Thank you for the question. Our audio encoder is trained in Stage 2 to capture the audio–motion mapping, so it cannot be replaced by other pre-trained audio feature extractors. In Stage 3, we fine-tune all parameters to refine the visual feature details and improve motion fidelity.

---

### Official Review · Reviewer_Kerd · 2025-10-31

**Soundness:** 3
**Presentation:** 2
**Contribution:** 2
**Rating:** 4
**Confidence:** 3

**Summary:**

This paper proposes a novel, weakly-supervised framework for co-speech gesture video generation that eliminates the need for pose annotations. Unlike previous two-stage methods that rely on explicit pose data, this approach learns an implicit motion representation directly from video. The framework has three key stages: a motion encoder learns motion representations from video without pose supervision; a dual-tower architecture with an invertible feature extractor maps audio directly into the learned motion space for inference; and a video diffusion model refines fine-grained visual details. It outperforms state-of-the-art methods (e.g., S2G, MYA) across multiple objective metrics (FGD, FVD, Diversity, BAS) and in user studies.

**Strengths:**

- I recognize this idea of dumping intermediate noise label supervision, which would make the training scalable. But I think this work does not work very well.
- The paper includes comprehensive experiments to evaluate the effectiveness of each design.

**Weaknesses:**

1. To support the motivation, the comparison should include the baseline trained with noise pose conditioned with the video model, either wan2.1 or other tiny model for efficiency. I do not think the current video models will perform  worse than the baselines included in the supplementary video.
2. The annotation error in in-the-wild videos always occurs in the hand region. However, most existing video models also fail to produce plausible details. Therefore, the work also employs an additional hand refinement model to enhance this. So, I think this kind of weak supervision and the motion encoder do not really work.
3. As observed in the ablation demo 1, the identities after applying each stage changed a lot. The gestures between stage 1 and stage 2 are totally different. This does not make sense if the audio encoder and motion encoder are  aligned well. I mean the poor semantics  alignment.
4. The work is limited by the constraints of current video diffusion models, which can only generate a few seconds of video, unlike explicit pose-driven methods.

**Questions:**

as listed in weekness

---

> ### Author Response · Authors · 2025-11-24
>
> Thank you for your acknowledgement of our core ideas and experimental evaluations.
>
> **W1: Comparisons with pose conditioned model**
>
> Thank you for the suggestion. The baselines we compare with, MYA and EchoMimicV2, are already pose-conditioned models. Since our task requires generating video directly from audio, these baselines cannot be used as it is: they require pose as input. To enable comparison, we employ the state-of-the-art audio-to-pose model DiffSheg to first generate poses, and then feed them into MYA or EchoMimicV2. This two-stage cascade inevitably accumulates errors, leading to the performance gap observed in the supplementary results.
>
> **W2: Effectiveness of our design**
>
> Thank you for the question. First, annotation errors in in-the-wild videos are not limited to the hands; they also appear in the face and upper-body regions. Using such noisy pose annotations as strong supervision can confuse the model, especially since our task does not require explicit pose control, and obtaining clean annotations at scale is costly.
>
> Second, our stage 3 already produces strong results, as shown in Table 2. The hand refinement module is optional and is used only to address a subset of cases with visible hand artifacts. In the video results, we deliberately included stage 3 samples with these artifacts to highlight the effect of the refinement. This does not imply that the motion encoder or weak supervision is ineffective. As shown in Table 2, our stage 1-3 design improves multiple metrics, including FGD, diversity, and BAS, demonstrating its overall effectiveness beyond just hand quality.
>
> **W3: Ablation study demo**
>
> Thank you for the comment. However, we do not observe the identity change or gesture drift between stages as described. Could you please clarify this? It is possible that this was an honest misunderstanding.
>
>
> **W4: Long video generation**
>
> Thank you for the comment. Unlike text-to-video generation, where the text provides limited control over appearance and long videos often suffer from drift, our model is an audio-controlled image-to-video generation framework, which maintains consistent appearance. Similar to other audio-driven video generation methods, it can produce long videos via clip-by-clip generation, using the final frames of the previous clip as the starting frames for the next. This approach is standard in audio-driven generation and is conceptually similar to techniques used in pose-driven methods.

---

### Official Review · Reviewer_UPPC · 2025-11-01

**Soundness:** 3
**Presentation:** 3
**Contribution:** 3
**Rating:** 8
**Confidence:** 4

**Summary:**

This paper proposes a novel weakly supervised framework for generating co-speech gesture videos directly from audio and video data, without requiring pose annotations. Traditional methods rely on two-stage pipelines (audio-to-pose, then pose-to-video), which demand extensive labeled pose data and often fail to capture fine-grained hand details.

The proposed framework consists of three major stages:

1. Motion Learning — A motion encoder learns generalizable motion representations from raw video data without explicit pose supervision.

2. Audio-Motion Mapping — A dual-tower architecture aligns audio and motion spaces using an invertible feature extractor based on normalizing flows, enabling efficient audio-to-motion mapping.

3. Feature Refinement — A video diffusion model refines visual details and ensures realism.

Additionally, the authors introduce a hand refinement strategy that optimizes the initial noise distribution during sampling using policy gradient methods.

Experiments on a self-constructed dataset (based on PATS) demonstrate that the proposed method achieves state-of-the-art performance across four key metrics—Fréchet Gesture Distance (FGD), Diversity, Beat Alignment Score (BAS), and Fréchet Video Distance (FVD)—outperforming prior works such as S2G, MYA, and EchoMimicV2.

**Strengths:**

The paper introduces a weakly supervised paradigm for co-speech gesture video generation that eliminates the dependence on explicit pose annotations—a major bottleneck in existing two-stage frameworks. This design is original and impactful, as it reframes gesture synthesis from a pose-supervised problem to a self-contained audio–video learning task. The incorporation of an invertible feature extractor for mapping between audio and motion spaces is also creative, offering an elegant solution that avoids retrieval-based inefficiencies or generative instability.

The proposed method is also carefully designed and systematically evaluated through multiple ablation studies that dissect each component’s contribution. Quantitative results show consistent and significant improvements across multiple metrics (FGD, BAS, FVD, and Diversity) compared to strong baselines such as S2G, MYA, and EchoMimicV2. The experiments also include both objective and user evaluations.

The paper is clearly written and well-structured. The motivation is compelling, situating the problem within the broader context of human–computer interaction and multimodal generation. The three-stage pipeline (motion learning, audio–motion mapping, and feature refinement) is logically presented and supported by clear figures and equations. The authors clearly distinguish their contributions from prior works such as ReenactAnything and S2G, making it easy to understand the conceptual leap from pose-supervised methods to weakly supervised motion learning.

Overall, the paper makes a meaningful advance in the field of gesture video generation and broader multimodal video synthesis. By removing the need for costly and error-prone pose annotations, the proposed framework substantially lowers the barrier to scaling audio-driven gesture datasets and models. This shift towards weak supervision could influence future research in gesture synthesis, human motion modeling, and audio-visual learning. The improved gesture realism and synchronization also have practical implications for digital avatars, virtual presenters, and embodied conversational agents.

**Weaknesses:**

1. The experiments are conducted solely on a small subset of the PATS dataset (33 hours) featuring four speakers. This raises concerns about the model’s ability to generalize to unseen identities, languages, or speaking styles. Since the proposed framework is claimed to learn motion representations without pose supervision, a more convincing demonstration would include cross-identity or cross-domain evaluations, such as applying the model to TED, Trinity, or BEAT datasets.
2. The paper does not discuss where the weakly supervised framework fails or underperforms compared to pose-supervised approaches. For instance, it remains unclear whether the learned motion encoder occasionally entangles motion and appearance, especially under large camera or body movements. Explicitly showcasing these failure cases and analyzing the underlying causes would make the paper more balanced and informative.
3. The framework involves multiple large components—VAE, 3D-UNet, invertible flow, and diffusion model—which may lead to high training and inference costs. However, the paper does not report computation time, GPU memory footprint, or efficiency comparisons with existing pose-supervised methods.

**Questions:**

1. Have the authors tested the framework on other gesture datasets such as BEAT, Trinity, or TED to assess cross-domain generalization?
2. How well does the model adapt to unseen identities or speaking styles without fine-tuning?
3. Could the authors visualize or analyze the learned motion embeddings (e.g., using t-SNE or PCA) to show that they capture meaningful motion dynamics rather than appearance cues?
4. What is the training time and GPU cost for each stage?
5. How does inference speed compare to previous supervised or two-stage baselines like S2G and MYA?

---

> ### Author Response · Authors · 2025-11-24
>
> Thank you for acknowledging our original design of removing pose supervision, the creative invertible mapping, the careful and systematic experiments with strong quantitative gains, as well as the clarity of our writing and motivation. We also appreciate your recognition of the work’s broader impact, future research potential, and practical relevance.
>
> **W1 & Q1 & Q2: Generalization**
>
> Thank you for your comments. As discussed in the related work, our co-speech gesture video generation task fundamentally differs from zero-shot audio-driven video generation. Zero-shot methods typically rely on datasets containing a large number of identities with only a few seconds of video per identity (e.g., Vlogger: 2.2k hours / 800k identities, ~9.9s per identity). Their goal is primarily lip synchronization with limited hand motion diversity.
>
> In contrast, our task aims to learn and reproduce the unique gesture style of a specific individual, which requires hours of video per identity. Therefore, unlike zero-shot settings, adapting to a new identity in our scenario necessarily involves fine-tuning, as the model must capture identity-specific gestural habits rather than generate generic talking-head motion.
>
> Regarding the datasets mentioned by the reviewer, we appreciate the suggestions. However, BEAT and Trinity do not provide videos and thus is incompatible with our method, which requires video-grounded motion supervision. TED contains only minutes of video per identity, which is insufficient for our task, as it cannot support learning personalized gesture styles.
>
> That said, when a new identity is available, a simple fine-tuning step on a few hours of that identity’s data is sufficient for our framework to adapt. This aligns with the core design goal of modeling individualized gesture behaviors rather than general cross-domain motion.
>
> **W2: Limitations**
>
> We acknowledge the reviewer’s concerns and appreciate the suggestion. Since our method follows the implicit motion extraction introduced in Reenact Anything (RA), our framework naturally inherits both its strengths and limitations. Regarding the potential appearance–motion entanglement, RA reports that pre-trained image-to-video models primarily derive appearance information from the latent image input, while the text/image embeddings injected through cross-attention mainly control motion. The same mechanism applies in our setting, where the output of our motion encoder is injected into the diffusion model via cross-attention, thereby mainly guiding motion.
>
> As for failure cases, we agree that the model performs less reliably under large camera/body movements. This is partly due to the limitation that the motion extraction relies on a pre-trained video generation model, which inherently struggles with large movements. Additionally, our current training set is predominantly composed of front-facing videos and does not include sufficient examples of large body movements or diverse camera dynamics.
>
> We will clarify these limitations in the revised manuscript and plan to address them in future work.
>
> Reenact Anything: Semantic Video Motion Transfer Using Motion-Textual Inversion. Kansy, Manuel, et al.  SIGGRAPH 2025.
>
> **Q3: Motion Embedding Analysis**
>
> Thank you for the suggestion. Our analysis reveals a separation between calm gestures and more energetic, movement-rich gestures, which tend to cluster independently. We will include this analysis in the revised manuscript.
>
>
> **W3 & Q4 & Q5: Efficiency Comparisons.**
>
> We thank the reviewer for raising these questions and apologize for the confusion. The training time and GPU cost for each stage are reported in Appendix B, and the inference speed and memory usage are provided in Appendix H. These results were placed in the appendix due to space constraints in the main paper.

---

### Author Response · Authors · 2025-12-03
**Rebuttal Summary for the AC**

Dear AC and reviewers,

We are grateful for the effort you have invested in reviewing our work. To support a clear and efficient evaluation, we summarize the our **contributions and strengths** below.

**1. The principal contributions introduced in our paper**

Our work introduces a novel weakly supervised framework for co-speech gesture video generation that removes the reliance on pose annotations—an important limitation of prior two-stage audio-to-pose and pose-to-video pipelines. Instead, we learn motion directly from raw audio–video data through a three-stage design.

First, we learn an implicit, generalizable motion representation from video using a motion encoder trained without pose supervision.
Second, we align audio and motion through a dual-tower architecture equipped with an invertible flow-based feature extractor, enabling efficient and direct audio-to-motion mapping.
Third, we employ a video diffusion model to refine visual details, with an additional policy-gradient-based hand refinement module that improves fine-grained hand realism by optimizing the diffusion noise distribution.

Across multiple metrics—including FGD, FVD, BAS, and Diversity—our method achieves state-of-the-art results, surpassing strong baselines such as S2G, MYA, and EchoMimicV2, and is further supported by user studies. Overall, our approach demonstrates that weak supervision can effectively replace pose annotations while offering improved gesture fidelity, diversity, and realism.

**2. The strengths consistently highlighted by the reviewers**

Original design of removal of pose supervision (UPPC)

Creative invertible audio–motion mapping design (UPPC)

Carefully designed and systematic experimental evaluation (UPPC, Kerd)

Strong quantitative improvements (UPPC)

High generation quality and solid performance (fVo4, ptuf)

Clear writing, compelling motivation, and well-presented method (UPPC, fVo4)

Meaningful broader impact, future research potential, and practical relevance (UPPC)

Strength of core ideas / method innovation (Kerd, fVo4, ptuf)


**In addition, please refer to the individual reviewer sections below for a detailed list of each reviewer’s concerns and our corresponding responses.**

---

### Meta-Review · Area_Chair_6AZ9 · 2025-12-31

**Summary:**

This paper proposes a weakly supervised learning method for gesture video generation in co-speech, a classic problem in digital humans. The authors address the unrealistic aspects of existing digital human generation by proposing three key components and provide qualitative and quantitative experiments for explanation and illustration.

This paper received one positive acceptance review and three negative reviews, leading to some controversy and discussion.

The area chair carefully reviewed the paper and the reviews, finding that the reviewers' comments focused on the dataset and evaluation methods, the reasonableness of the baseline evaluation model, and the depth of theoretical exploration. Furthermore, the actual video performance did not receive sufficient approval from the reviewers, including the effectiveness of duration and motion. Specific comments are summarized below.

**Reviewer Concerns:**

+ The reviewer UPPC raised further concerns regarding the dataset size, generalization ability, and training and inference costs. The authors stated that the research characteristics of this paper differ significantly from zero-shot testing, and other datasets were incompatible, making evaluation impossible. In addition, regarding the exploration of limitations and the underlying reasons, the authors stated that they mainly follow the work of Kansy, Manuel et al. 2025. Further analysis and presentation will follow in future work. AC believes these responses partially address the reviewers' comments, but there is still significant room for improvement regarding concerns about experimental generalization and the depth of theoretical analysis.

+ Reviewer Kerd expressed concerns about the baseline conditions for video model training, including annotation errors shown in the demo and Seagate's concerns. They also expressed concern about poor semantic alignment in ablation analysis. The authors acknowledged some annotation errors but acknowledged some improvements. Regarding long video analysis and generation, the authors stated that it could be addressed through frame-by-frame generation, but the area chair and reviewers could not draw more credible conclusions.

+ Reviewer fVo4 expressed concerns about the motivation for not requiring pose annotations, the performance of fine-tuning the CLIP algorithm for video reconstruction, and the purpose of dynamic transformation. They also raised some questions about ablation analysis. The authors provided a description of the uniqueness of their task and emphasized their research characteristics and contributions. Furthermore, regarding other settings, the authors largely follow existing work, which they believe is reasonable to some extent. The area chair felt these responses still lacked sufficient evidence, particularly in numerical analysis, visualization results, theoretical analysis, and proof. These may require further revision and supplementation.

+ Reviewer ptuf expressed concerns about the methodology, particularly the lack of reasonable motivation to integrate these viewpoints, including many of the proposed modules. Additionally, the reviewer listed more related work for comparative analysis. Furthermore, they expressed further concerns regarding visualization in video generation and generation duration.

**Reviewer Scores:**

At this stage, the authors clarified some details and conducted technical and methodological analyses with some related work, but did not provide further evidence to support their conclusions, especially in the visualization analysis of video demos, etc.

In summary, this paper received one positive review and 3 negative reviews, but all reviewers expressed concerns about the experimental setup, visualization, settings, and experimental comparisons. Some reviewers also raised concerns about the method's innovativeness. The authors clarified some details at the rebuttal stage, but there were not enough revisions to convince or support all reviewers' conclusions. The AC carefully read all comments and the authors' responses, and accepted the reviewers' unanimous conclusion that the paper received a negative score and needed further revisions before being submitted to the next meeting.

---

### Decision · Program_Chairs · 2026-01-26

Reject